# *In Vitro* Rumen Fermentation and Post-Ruminal Digestibility of Sorghum–Soybean Forage as Affected by Ensiling Length, Storage Temperature, and Its Interactions with Crude Protein Levels

**DOI:** 10.3390/ani12233400

**Published:** 2022-12-02

**Authors:** Temitope Alex Aloba, Uta Dickhoefer, Joaquin Castro-Montoya

**Affiliations:** 1Institute of Animal Nutrition and Rangeland Management in the Tropics and Subtropics, University of Hohenheim, 70599 Stuttgart, Germany; 2Institute of Animal Nutrition and Physiology, Christian-Albrechts Universität zu Kiel, 24118 Kiel, Germany; 3Graduate School, Faculty of Agricultural Sciences, University of El Salvador, San Salvador, El Salvador

**Keywords:** ensiling, temperature, crude protein, rumen fermentation, intestinal digestibility

## Abstract

**Simple Summary:**

The purpose of the study was to evaluate the effects of ensiling length, storage temperature, and its interaction with crude protein (CP) levels in sorghum-soybean forage mixtures on in vitro rumen fermentation and post-ruminal nutrient digestibility. Ensiling until 75 day (d) increased the microbial end products of rumen fermentation in comparison to fresh forage and to forage ensiled beyond 75 d. The outdoor storage temperature of ensiled sorghum-soybean forage influenced the post-ruminal digestibility of CP negatively, whereas increased CP levels positively influenced rumen fermentation and post-ruminal digestibility. In conclusion, ensiling beyond 75 d reduces CP digestibility significantly.

**Abstract:**

The study aimed to evaluate the effects of ensiling length, storage temperature, and its interaction with crude protein (CP) levels in sorghum–soybean forage mixtures on in vitro rumen fermentation and post-ruminal digestibility of nutrients. The dietary treatments consisted of fresh forages (d 0) and silages of sorghum and soybean stored indoors or outdoors for 75 and 180 d with additional ingredients to make two dietary CP levels, 90 and 130 g/kg dry matter (DM) and a forage-to-concentrate ratio of 80 to 20. An in vitro procedure was conducted using the ANKOM RF technique to study rumen fermentation. The dietary treatments were incubated in duplicate for 8 and 24 h in three runs. After each incubation time, in vitro rumen fermentation parameters were measured, and the protozoa population was counted using a microscope. Post-ruminal digestibility was determined using the pepsin and pancreatic solubility procedure. Cumulative gas production (GP) increased quadratically with ensiling length (8 h, *p* < 0.01; 24 h, *p =* 0.02), and the GP differed between CP levels at both incubation times (*p* < 0.01). However, total short-chain fatty acid (SCFA) concentrations in rumen inoculum increased quadratically with ensiling length (*p* < 0.01; for both incubation times), and interaction between ensiling length and CP levels was observed in proportions of acetate and propionate after 24 h of incubation (*p* < 0.01; for both incubation times). Similarly, an interaction between ensiling length and CP levels was found in the proportion of valerate after 24 h of incubation (*p* < 0.01). There was a quadratic response to ensiling length in the NH_4_–N concentration after 8 h (*p* < 0.01) and 24 h (*p* < 0.05), and the CP level also differed (*p* < 0.01) at both incubation times. The ciliate protozoa count after 24 h was higher in low CP diets than in high CP diets (*p* = 0.04). The amount of CP in the undegraded substrate at both incubation times differed between CP levels (*p* < 0.01; for both incubation times). An interaction effect between ensiling length and storage temperature after 8 h (*p* = 0.02) and 24 h (*p* < 0.01) was observed for intestinal CP digestibility. The effect of CP levels on intestinal CP digestibility differed after 8 h (*p* < 0.01) and 24 h (*p* < 0.01). In conclusion, increasing ensiling length beyond 75 d reduced CP digestibility, and additional CP inclusion did not ameliorate this.

## 1. Introduction

The microbial degradation of carbohydrates and crude protein (CP) in the rumen by diverse microbes with an expected net feeding strategy is a complex process that takes place before nutrient absorption post-ruminally. To assess the nutritional value of dietary CP in ruminants, the amount of undegraded CP in the rumen that flows to the duodenum, its intestinal digestibility, and the amino acid composition of the undegraded CP is required [1,2]. The flow of rumen undegraded CP post-ruminally may vary among forage species and between fresh forages and silages [3]. Moreover, a greater flow of undegraded CP does not always lead to increased amino acid absorption [4]. For instance, Cone et al. [5] reported an increased rumen undegraded CP concentration and a reduced intestinal digestibility of rumen undegraded CP for *Lolium perenne* grass. In contrast, a decrease in rumen undegraded CP concentration and increased intestinal digestibility of rumen undegraded CP was found for *Lolium perenne* grass silage using in situ and in vitro methods. In another study by Lima and colleagues [6], the digestible CP supply in the small intestine of sheep fed fresh sorghum–soybean was lower than the intestinal digestible CP supply for ensiled sorghum–soybean forage.

The differences in the growth habits and morphological characteristics of diverse tropical forage legumes showed differences in the CP digestibility of tropical forage legumes in ruminants [7]. Moreover, the protein quality and the CP fractions of tropical forage legumes that differ between species changed during conservation techniques [8,9]. In view of the impacts of ensiling conditions, such as a prolonged ensiling period and storage temperature, the variation in the intestinal CP digestibility between fresh and silage of forage mixtures may be amplified. For instance, ensiling three different tropical forage legumes and their combinations with sorghum over a prolonged period of 180 d in a high ambient temperature increased the proportion of acid detergent insoluble nitrogen (ADIN), decreased the neutral detergent insoluble nitrogen (NDIN), and slowly degraded the CP fraction [8]. 

In consideration of the preceding, changes in the CP of ensiled forages are imminent, and the CP supply from ensiled legumes for rumen microbes and post-ruminal use may be limiting. Diets containing tropical forage legume silage resulted in higher performance (average daily gain, milk yield) than those without tropical legume silage inclusion, and it is assumed that forage legume CP appeared to be more digestible post-ruminally [9]. Thus, evaluating how diets with ensiled forage legumes behave under low and high (relatively moderate) CP conditions with the premise that if CP from ensiled forage legumes is limiting, then under low CP conditions, negative effects would be exacerbated. Similarly relevant is knowing how ensiling conditions affect the contribution of CP post-ruminally since it is generally assumed that silage’s CP are extensively degraded in the rumen [10] and that the intestinal digestibility of undegraded CP cannot be assumed to be constant for a particular feed [11]. Thus, the present study evaluated the in vitro rumen fermentation and post-ruminal digestibility of diets containing fresh or ensiled sorghum and soybean forage combination as affected by the ensiling length, storage temperature, and its interaction with dietary CP levels. 

## 2. Materials and Methods

### 2.1. Experimental Diets

In the present in vitro study, three factors were studied: the effect of ensiling length (i.e., 0, 75, and 180 days), the impact of storage temperature (i.e., indoor vs. outdoor) and the effect of dietary crude protein levels (high vs. low). Sorghum and soybean forage samples ensiled under different storage temperatures and lengths from a previous silage experiment [8] were used as basal feeds mixed with other ingredients to form the diet. Details on the agronomic practices, ensiling conditions, and the chemical and fermentation characteristics of the forage samples used in the current study have been described in a previous silage experiment. The average storage temperature observed over the ensiling period for silages stored indoors and outdoors is 25 °C and 30 °C, respectively. Diets were formulated at a constant forage-to-concentrate ratio of 80 to 20 (on a dry matter (DM) basis), with sorghum forage or silage representing 48% and soybean forage or silage 32% of the diets (on a DM basis). The concentrate mixture in the diets comprised corn starch (9444.1, Roth GmbH, Karlsruhe, Germany) and soy protein (066-974, ProFam^®^ 974 ADM, Decatur, IL, USA) in different proportions to achieve diets with two CP concentrations (low CP; 90 g/kg DM and high CP; 130 g/kg DM). In addition to the fresh forage diet at 2 levels of CP concentration, 8 dietary treatments were tested (2 storage temperature × 2 ensiling lengths × 2 dietary CP concentrations).

### 2.2. In Vitro Fermentation with the ANKOM RF Technique

The in vitro experiment was conducted using an ANKOM RF gas production system (ANKOM Technology, Macedon, NY, USA) equipped with 22 units, which releases the accumulated gas automatically in the flask headspace at 0.7 psi pressure through an ANKOM sensor module. 

Before the morning feeding, rumen fluid (2.9 L/incubation) was collected from various locations within the rumen of three rumen-cannulated dry Jersey cows using a perforated hose attached to a vacuum pump. All cows had free access to fresh drinking water and were fed ad libitum a total mixed ration composed of (per kg DM) corn silage (329 g), grass silage (329 g), grass hay (229 g), barley straw (100 g), urea (5 g), and a mineral mixture (8 g: 0.4 g calcium, 1.3 g phosphorus, 1.4 g magnesium, and 4.9 g sodium).

Rumen fluid was transported to the lab in a pre-warmed, insulated flask and strained through a gauze bag of 100-μm-pore size. The strained rumen fluid was mixed with a preheated (39 °C) standard buffer solution according to Menke and Steingass [12] under constant stirring and continuous flushing with carbon dioxide in a water bath (39 °C). 

Each substrate ingredient (480 mg of sorghum, 320 mg of soybean, and 200 mg of corn starch or soy protein + corn starch) was weighed separately into 500 mL Duran bottles for every run to compose a total of 2 g of mixed substrate. Subsequently, 300 mL of rumen inoculum were added to each Duran bottle, its headspace saturated with carbon dioxide, sealed, and placed in a water bath at 39 °C for 8 h and 24 h incubation periods. Within each run, each experimental diet was incubated in duplicate per incubation period. Additionally, two blank bottles per incubation time with only rumen inoculum were included in each run to correct gas production (GP), total short-chain fatty acid (SCFA) concentration, and apparent degradability.

### 2.3. Sampling

At the end of each incubation period, GP measurement was recorded. Additionally, the incubation medium’s pH was recorded using a pH-meter (WTW Multi 340 i, WTW, Weilheim, Germany). Then, an aliquot of 750 μL of incubation medium was taken for protozoa count. The aliquot was fixated with 750 μL of methyl green formalin-saline solution (10 mL formaldehyde solution (35%, v/v); 90 mL distilled water; 0.06 g methyl green; 0.8 g sodium chloride) and stored at 4 °C in a refrigerator until counting. 

Afterwards, the remaining contents of each Duran bottle were transferred to polyethene bottles and centrifuged at 500× *g* at 4 °C for 10 min (Hettich Rotanta, Tuttlingen, Germany). Two aliquots of 5 mL of decanted supernatant were collected and stored at −20 °C for determination of SCFA and ammonium–nitrogen concentrations. After centrifugation and decantation of the supernatant, the residual pellet (in vitro apparently degraded DM) was obtained, lyophilized, weighed, and ground it using a ball mill (Retsch, MM200, Haan, Germany) for 2 min at a frequency of 30 s, and then stored at room temperature until the determination of in vitro intestinal digestibility.

### 2.4. Chemical Analysis

The DM, crude ash, and ether extract concentrations of each ingredient were determined according to the official analytical method in Germany (VDLUFA) [13] in duplicates. Nitrogen (N) was analyzed by Dumas combustion using a Vario MAX CN element analyzer (Elementar Analysensyteme GmbH, Hanau, Germany) to determine CP concentrations (CP = N × 6.25) concentration (method 4.1.1 of VDLUFA). Similarly, N concentrations in residual pellets were also determined using a CN analyzer. The concentrations of neutral detergent fiber (aNDF; assayed with heat-stable amylase and sodium sulphite) and acid detergent fiber (ADF) were analyzed in sequence using the ANKOM 200 fiber analyzer (ANKOM Technology, Macedon, NY, USA) (methods 6.5.1 and 6.5.2 of VDLUFA). Each substrate’s nutrient concentrations were then calculated from the chemical composition of each ingredient in duplicate (Table 1 and Table 2).

For SCFA analysis, 2 mL of each aliquot of the supernatant obtained from initial centrifugation was transferred into vials and later centrifuged at 20,000× *g* at 4 °C for 10 min (Avanti™ 30, Beckman Coulter™, Indianapolis, IN, USA). An aliquot of 720 μL of the supernatant of this centrifugation was pipetted into a 1.5 mL vial, mixed with 80 μL of an internal standard (1 mL methyl valeric acid dissolved in 99 mL formic acid), and stored at 4 °C to precipitate the soluble proteins [14]. Following this, the mixture was centrifuged at 20,000× *g* (10 min, 4 °C), and 800 μL of the supernatant was transferred into 1.5 mL glass vials before analyzing for SCFA by a gas chromatograph (GC 14-A Shimadzu Corp., Kyoto, Japan) equipped with an auto-injector (AOC–20i, Shimadzu Corp., Kyoto, Japan).

Ammonium–nitrogen (NH_4_–N) concentration was determined in duplicate according to [15]. For this, an aliquot of 20 μL of the supernatant obtained after the first step centrifugation for SCFA analysis was pipetted into a 2 mL vial with the addition of 900 μL of reagent A (2.5 g phenol hypochlorite and 12.5 mg sodium-nitroprusside dissolved in 250 mL distilled water). Subsequently, the mixture was centrifuged at 10,000× *g* for 10 min at 4 °C (Biofuge, Heraeus Holding GmbH, Hanau, Germany). Reagent B (900 μL; 2.5 g sodium hydroxide + 2.1 mL sodium hypochlorite (containing 12% (v/v) chlorine)) was then added after 4 min, and the mixture incubated at 38 °C for 20 min. After incubation, the solution was transferred to a cuvette, and the NH_4_–N concentration was read at 625 nm using a spectrophotometer (Varian Cary 50 Bio, UV–vis, Palo Alto, CA, USA).

The method of Boisen and Fernández [16] modified by Westreicher-Kristen et al. [17] using the pepsin and pancreatic solubility procedure (PPS) was adopted to determine the in vitro intestinal digestibility of DM and CP. For this, the residual pellets obtained after incubation were pooled per experimental diet and incubation period. For each incubation run of the PPS analysis, pooled samples were analyzed in triplicate simultaneously with two blanks containing only incubation medium to correct for the PPS. Pooled samples (400 mg) suspended in a 100 mL conical flask were thoroughly mixed with 25 mL phosphate buffer (0.1 M; pH 6.0). To the mixture, 10 mL of 0.2 M hydrochloric acid was added, and its pH was adjusted to 2 using 1 M sodium hydroxide or 1 M hydrochloric acid. Then, 1 mL of pepsin solution (0.01 g/mL; Merck 7190, 200 FIP U/g) was added to the mixture before it was incubated at 40 °C in an oven for 6 h under constant stirring. Afterwards, 5 mL of 0.6 M sodium hydroxide and 10 mL of phosphate buffer were added, and the pH of the samples was adjusted to 6.8 with 5 M HCl or 5 M sodium hydroxide. Subsequently, 1 mL of freshly prepared pancreatin solution (0.05 g/mL; Sigma P-1750, Sigma–Aldrich, Burlington, MA, USA) was added, and the mixture was incubated in an oven at 40 °C for 18 h under constant stirring. After incubation, 5 mL of 20% (v/v) of sulfosalicylic acid solution was added to the incubated mixture, which was then left to stand at room temperature for 30 min. Then, the entire contents of the flasks were filtered through a previously weighed filter paper (Whatman paper N° 54, GE Healthcare Life Sciences, Darmstadt, Germany) that was oven-dried at 103 °C for 2 h. The insoluble residue in the filter paper was washed with ethanol and acetone, oven-dried again at 103 °C for 4 h, and weighed. This insoluble residue was considered to be the apparent in vitro intestinal undigested DM. Finally, the N concentration in this residue was determined by Kjeldahl to calculate the intestinally undigested CP.

The ADIN concentrations of the diets were determined following the standardization procedures for N fractionation [18]. 

### 2.5. Protozoa Count

For ciliate protozoa count, 1 mL each of the fixated samples was pipetted into two Fuchs–Rosenthal chambers (0.2 mm depth, 2 × 2 mm chamber, 0.25 mm square lined), and ciliate protozoa were counted under 10 × magnification in a microscope (Zeiss, Carl Zeiss Microscopy GmbH, Jena, Germany). The total number of protozoa per mL of fixated sample was calculated from the average counts of the two chambers, and the protozoa count of incubated blank samples was used for correcting the dietary treatment. 

### 2.6. Calculations

The non-structural carbohydrate (NSC) concentration was calculated according to the equation of NRC [19] as follows:NSC = 1000 − (ash + CP + CL + aNDF)(1)
with NSC, ash, CP, CL and aNDF in g/kg DM

The metabolizable energy (ME) of basal ingredients for each diet was estimated using the GfE [20] equation, and the concentrations of crude nutrients, cumulative gas production (GP), and ADFom (method 6.5.2 of VDLUFA) for the ME equation were obtained from the previous silage experiment [8].
ME = 12.49 − (0.0114 × ADFom) + (0.00425 × CP) + (0.0269 × CL) + (0.01683 × GP)(2)
with ME in MJ/kg OM; CP, CL, and ADFom in g/kg OM; and GP in mL/200 mg OM.
ME (MJ/kg DM) = ME (MJ/kg OM) × (1000 − CA (g/kg DM))/1000(3)

The ME concentration of corn starch was obtained from Schiemann et al. [21] and soy protein from Van Eys et al. [22].

The in vitro apparent ruminal DM degradability was calculated as the difference between the substrate DM and the residual dry mass corrected for the residual dry mass from the blanks after in vitro incubation and divided by the substrate DM, expressed in percentage. The in vitro apparent undegraded CP was calculated from the CP concentrations in the residual substrate after fermentation, corrected for CP concentration in residual substrate recovered from the blanks. It was assumed that all N determined in the residue originated only from undegraded substrate CP. However, proportions of undegraded CP were likely overestimated due to the attachment of microbial matter to the residual substrate and the contribution of N from the buffer solution. For in vitro apparent intestinal DM digestibility calculation, the dried residual pellets corrected for the blank residual dry mass after PPS incubation was subtracted from the residual dry mass after ANKOM incubation and divided by the residual dry mass (ANKOM incubation) expressed in percentage. The in vitro apparent total DM digestibility was calculated by multiplying the corrected residual dry mass after ANKOM incubation by the apparent intestinal digestibility coefficient and summed with the difference between the substrate DM and the corrected residual dry mass after ANKOM incubation and then divided by the substrate DM expressed in percentage. In vitro apparent intestinal CP digestibility was calculated as the difference between the apparent undegraded CP concentration after ANKOM incubation and the residual CP concentration after PPS incubation divided by the apparent undegraded CP concentration after ANKOM incubation expressed in percentage.

### 2.7. Statistical Analysis

Data were analyzed using the mixed model (PROC GLIMMIX) of SAS 9.4 (SAS Institute, Inc., Cary, NC, USA). The main effect of ensiling length, storage temperature, CP of diets, and their interactions for each incubation period at different sampling hours (n = 6, 2 duplicates × 3 incubations) was analyzed according to the model:Y_ijk_ = µ + L_i_ + T_j_ + P_k_ + (LT)_ij_ + (LP)_ik_ + (TP)_jk_ + (LTP)_ijk_ +e_ijk_
where Y_ijk_ = dependent variable, µ = overall mean, L_i_ = ensiling length effect, T_j_ = storage temperature effect, P_k_ = crude protein level effect, (LT)_ij_ = the interaction effect of ensiling length and storage temperature, (LP)_ik_ = the interaction effect of ensiling length and crude protein level, (LTP)_ijk_ = the interaction effect of ensiling length, storage temperature and crude protein level, and e_ijk_ = residual random error of experiment.

For cases in which no interaction effect was observed for any variable, those were not reported. Linear and quadratic effects of ensiling length were determined using orthogonal polynomial contrasts. All significant differences were declared at *p* < 0.05.

## 3. Results

### 3.1. Fermentation Parameters

There was no effect of storage temperature on in vitro rumen fermentation parameters. In addition, no interaction was found between storage temperature and ensiling length, storage temperature and crude protein level, and the interaction of the three studied factors. Irrespective of the incubation time, GP was influenced quadratically by ensiling length (8 h, *p* < 0.01; 24 h, *p =* 0.02; Table 3) with increase in GP from 0 to 75 d of ensiling before declining at 180 d. The GP was higher in diets with low rather than high CP concentration at both incubation times (*p* < 0.01; for both incubation times). Additionally, the pH was greater in diets with high rather than low CP concentration after 24 h incubation (*p* < 0.01). 

The NH_4_–N concentrations in the inoculum were greater for all diets after 24 h than after 8 h incubation. An interaction effect between ensiling length and CP level was found for NH_3_–N concentrations after 8 h incubation (*p* = 0.04), with greater NH_3_–N concentrations for high rather than low CP diets in all ensiling lengths and with greater absolute difference between CP levels at day 0 than 75 and 180 d. A quadratic response with increasing ensiling length was found for NH_4_–N concentration after 24 h (*p* < 0.05) of incubation, increasing the NH_4_–N concentration from 0 to 75 d and declining at 180 d. Moreover, after 24 h incubation, the NH_4_–N concentration was greater for high rather than low CP diets (*p* < 0.01). 

Counts of ciliate protozoa decreased with advancing ensiling length quadratically (*p =* 0.02) after 8 h incubation with the highest and lowest ciliate protozoa counts at 0 and 75 d, respectively. Additionally, there was a linear decrease with increasing ensiling length in counts of ciliate protozoa after 24 h of incubation (*p* < 0.01). The protozoa counts were greater in diets with low rather than high CP concentration (*p* = 0.04) after 24 h of incubation. 

Quadratic responses to prolonging ensiling length were found for total SCFA concentration after 8 h and 24 h of incubation (*p* < 0.01, for both incubation times; Table 4), with an increase in total SCFA concentration from 0 to 75 d of ensiling before declining at 180 d. Similarly, a quadratic effect to increasing ensiling length was found for the acetate proportion, increasing from 0 to 75 d of ensiling before declining at 180 d (*p* < 0.01). Acetate proportion was greater for high rather than low CP diets after 8 h of incubation. However, after 24 h of incubation, an interaction effect was found between ensiling length and CP level for the acetate proportion (*p* < 0.05), with a greater acetate proportion for low rather than high CP diets in all ensiling lengths and with a greater absolute difference between CP levels at 180 rather than 75 and 0 d ensiling lengths. There was also an interaction between ensiling length and CP level for the propionate proportion after 8 h (*p* < 0.05) and 24 h (*p* < 0.01) of incubation. 

While the propionate proportion was not affected by the CP level at day 0 of ensiling, the propionate proportion was higher in the high CP diets at 75 and 180 d. 

A linear increase in the isobutyrate proportion with prolonged ensiling length (*p* < 0.05) was found, and the proportion of isobutyrate was greater for low rather than high CP diets after 8 h of incubation. However, after 24 h of incubation, the isobutyrate proportion increased with advancing ensiling length quadratically (*p* < 0.05), with the highest and lowest propionate proportion at 75 and 0 d ensiling length, respectively. An interaction effect between ensiling length and CP level was found for the butyrate proportion (*p* < 0.01) after 8 h of incubation, with a greater butyrate proportion for low rather than high CP diets and greater total difference between CP levels at 75 and 180 rather than 0 d ensiling length. The isovalerate proportion increased linearly with ensiling length (*p* < 0.01), and the proportion of isovalerate was lower in high rather than low CP diets after 8 h of incubation. Moreover, after 8 h of incubation, the valerate proportion increased with increasing ensiling length (*p* < 0.01), with a greater proportion of valerate for silages stored indoors rather than outdoor (*p* < 0.05) and a higher proportion for high rather than low CP diets (*p* < 0.01). 

Additionally, an interaction effect between ensiling length and CP was found for the proportion of valerate after 24 h of incubation (*p* < 0.01), with a higher valerate proportion for low rather than high CP diets and with a greater absolute difference between CP levels at 75 rather than 180 and 0 d ensiling lengths. The proportion of acetate to propionate after 8 h of incubation decreased quadratically with advancing ensiling length (*p* < 0.01), with the highest and lowest proportion at 0 and 75 d ensiling length, and a greater proportion of acetate to propionate was found in silages stored outdoors rather than indoors (*p* < 0.05). Nevertheless, after 24 h of incubation, an interaction effect between ensiling length and CP levels was found for the proportion of acetate to propionate, with a greater proportion for low rather than high CP diets and a greater absolute difference between CP levels at 75 rather than 180 and 0 d ensiling length. The proportion of branched chain fatty acids (BCFA) increased linearly with ensiling length after 8 h (*p* < 0.01) and 24 h (*p* < 0.05) of incubation, and the CP differed after 8 h of incubation, with a greater proportion of BCFA found for low rather than high CP diets.

### 3.2. In Vitro Rumen Degradability and Post-Ruminal Digestibility

There was no effect of all studied factors on the apparent rumen DM degradability. A quadratic response of apparent intestinal DM digestibility with increasing ensiling length was found after 8 h of incubation (*p* < 0.01 Table 5), whereas it decreased linearly after 24 h of incubation (*p* < 0.01). Similarly, the apparent total DM digestibility after 24 h of incubation decreased linearly (*p* < 0.01) with increasing the ensiling length. 

The CP concentration in the undegraded substrate after 8 h and 24 h of incubation (*p* < 0.01; for both incubation times) decreased linearly with advancing ensiling length. Additionally, the amount of CP concentration in the undegraded substrate was greater in high rather than low CP diets after 8 h and 24 h of incubation (*p* < 0.01 for both incubation times).

There was an interaction effect between ensiling length and storage temperature for apparent intestinal CP digestibility after 8 h (*p* = 0.02) and 24 h (*p* < 0.01) of incubation. The apparent intestinal CP digestibility was greater for silages stored outdoors rather than indoors at 75 d and lower for silages stored outdoors rather than indoors at 180 d after 8 h of incubation and with no difference between storage temperature at 75 d and 180 d. Additionally, the apparent intestinal CP digestibility was greater for indoor rather than outdoor storage at 75 d and 180 d after 24 h of incubation and with a greater absolute difference between storage temperatures at 180 d than 75 d. Moreover, after 8 h and 24 h of incubation, apparent intestinal CP digestibility was greater (*p* < 0.01 for both incubation times) in diets with high rather than with low CP concentration.

## 4. Discussion

### 4.1. In Vitro Rumen Fermentation

The anaerobic microbial fermentation end products in the rumen are gases, methane, ammonia, and, most importantly, SCFA, which provides ruminants with a major source of metabolizable energy, and is considerably influenced by diet [23]. Increasing the CP level of diets reduced the GP, and the GP response to the increasing ensiling length was quadratic, with the highest GP at around 75 d of ensiling length. Previous studies [24,25,26] have shown that the contribution of protein fermentation to GP is negligible compared to carbohydrate fermentation, which is consistent with the finding in the present study. The greater concentration of NSC in low CP diets and in diets from forages ensiled at 75 d that were rapidly fermented likely enhanced fiber degradation, thereby increasing the GP. Although no differences were found across diets for DM degradability in the present study, it appears that energy supply and availability of ruminal N increased microbial growth and activities, which positively influenced GP [25,27,28]. Additionally, the greater GP from diets with silages stored for 75 d compared to those ensiled for 180 d and as compared to the diets from fresh forages may be associated with the greater availability of ruminal N that promoted rumen microbe production, thereby increasing fiber degradation. Moreover, the fiber concentrations of silages stored for 75 d may be more degradable, indicating lower usage of readily fermentable carbohydrates by silage microbes during ensiling at 75 d compared to 180 d. 

Accordingly, the total SCFA concentrations in rumen inoculum increased quadratically with increasing ensiling length and with a tendency for the effect of CP level. The tendency for higher total SCFA concentrations with a low CP diet is consistent with the high GP and the decrease in the rumen inoculum pH. This might be attributable to the greater NSC concentration and greater digestion supplying a more fermentable substrate for rumen fermentation. Additionally, the action of silage inoculant on preserving NSC of forages ensiled at 75 d rather than 180 d could have enhanced fiber accessibility by rumen microbes better than other ensiling lengths [3], thereby resulting in greater total SCFA concentrations, as some studies have reported improvement in DM and fiber digestibility of silage treated with mixed bacterial inoculant [29,30], such as was used during ensiling in the previous silage study [8]. The higher fiber and ADIN concentration in forages ensiled at 180 d rather than 75 d as associated with the reduction of soluble carbohydrates during ensiling [8] could be attributed to the decline of total SCFA concentration in the rumen with diets from forages ensiled for 180 d.

Moreover, the interaction of ensiling length and CP levels stimulated varying shifts in the profile of individual SCFA proportions in rumen inoculum. The interaction between ensiling lengths and CP levels is a reflection of the higher ratio of NSC to ADF concentration in the diets from forages ensiled at 75 d, as it influences the shift in the ratio of acetate to propionate. Additionally, the supplemented CP in high CP diets provided rumen fermentation with additional hydrogen sinks, thereby increasing the propionate level. 

The relatively high proportion of propionate in the rumen inoculum with increasing ensiling lengths agrees with other studies showing that lactate in silages is predominantly fermented into propionate in the rumen [31,32,33]. Additionally, the enhanced propionate proportion in high CP diets is likely related to the higher proportion of grain ingredients in high CP diets, which have typically been reported to increase the propionate proportion [34]. 

Butyrate is primarily produced by protozoa [35], as consistent with higher protozoal counts in inoculum from diets with lower CP concentrations in the present study. Equally, there is a positive correlation between protozoal populations and increased starch concentration [36], and this was observable in the current study. Furthermore, starch is an essential substrate to protozoa [35]. On the contrary, holotrich protozoa have limited ability to degrade structural carbohydrates [37]. These protozoa species may have constituted most protozoa counted in the present study, which reduces with higher fiber concentration related to increasing ensiling length. 

### 4.2. Ruminal Degradation and Post-Ruminal Digestibility

There was no difference across diets for ruminal DM degradation, but an increase in DM degradation with increasing incubation time was observed. The increase in the availability of N for microbial growth and the time provided for rumen microbes to attach and degrade diets may be responsible for the increased DM degradation at 24 h of incubation. However, there was a linear decrease in the digestibility of undegraded DM post-ruminally with increasing ensiling length. This indicates that with increasing ensiling length, the fiber component of the diets was less digested by rumen microbes.

Primarily, ruminal CP degradation is affected by protein solubility, interaction with other nutrients, and the predominant microbial population [4]. The NH_4_–N concentrations in rumen inoculum were greater with high rather than low CP concentration for all diets and at both incubation times. This observation is due to the greater CP concentration for high rather than low CP diets as soy protein was included in the high CP diet composition, suggesting that dietary CP level plays a major role in ruminal protein degradation. Similarly, Dung et al. [38] found a greater NH_4_–N concentration in rumen inoculum with increasing dietary CP from 100 g to 190 g/kg DM. Moreover, the NH_4_–N concentrations in rumen inoculum increased quadratically with advancing ensiling length, showing greater CP degradation. The variation between the proportion of true protein in silages at 75 d and 180 d that originated from the increase in protein degradation to NH_3_ during ensiling with increasing ensiling length in our previous study [8] may be related to this quadratic response. 

The concentration of CP in the undegraded substrate decreased linearly with increasing length, suggesting higher substrate CP degradation from silage diets than fresh forage diets. This observation may be explained by the increased soluble CP in silages, consistent with findings from previous studies on silages [10,32,39]. Additionally, the rate of degradation of non-protein nitrogen and soluble CP from silages in the rumen is high [10], and this was reflected in the higher proportion of valerate and concentration of ruminal NH_4_–N produced from silage diets rather than from fresh forage diets in the present study. Overall, the likely overestimation of the amount of CP in the undegraded substrate in the current study might be due to the rumen fluid’s microbial mass nitrogen contribution. 

The fiber-bound protein proportion in both soybean and sorghum silages increased with ensiling length and was greater for outdoor storage temperatures than indoor during ensiling in our previous study [8]. Therefore, the decline in apparent intestinal CP digestibility with increasing ensiling length and in outdoor storage may be related to the considerable reduction in the soluble CP fraction, leading to the increase in the proportion of the indigestible CP fraction in the total CP of particulate matter [5]. It is well established that the slowly degraded CP fraction (B_3_) of feed escapes ruminal degradation, thereby making it available for digestion in the lower gut [18]. Accordingly, the decline in the apparent intestinal CP digestibility with increasing ensiling length may be related to the decrease in the proportion of B_3_ that was mediated by the rise in the the ADIN proportion of the silages stored outdoors with increasing ensiling length from our previous study, considering that the proportion of forage in the diet contributed more to the indigestible CP fraction.

Lima and colleagues [6] observed a positive effect of ensiling on sorghum–soybean forage mixtures. In that study, forage ensiled between 162–182 d showed a higher proportion of intestinal digestible CP, although the authors did not provide details of the causal factors. Additionally, the apparent intestinal CP digestibility increased with diet CP concentration in the present study. Previous studies have reported an apparent intestinal CP digestibility of 98% for soy protein using the modified three-step procedure [40], higher than the 93% assumption of the NRC [19] model. Therefore, the proportion of soy protein in the diet with high CP concentration might have contributed more to the increase in the intestinal CP digestibility than the forage or silage proportion, apart from the likely overestimation of the intestinal CP digestibility due to the rumen fluid’s microbial mass nitrogen contributions. Although correction for microbial CP for substrate residues after the in vitro incubation to quantify the CP contribution of microbial origin was not done, the contribution of the silage CP proportion in the residue might be low, given that the acid detergent insoluble nitrogen (ADIN) and ADF concentration of sorghum and soybean silage are higher than that of soy protein.

## 5. Conclusions

The results of this study demonstrate that ensiling length had a greater impact on silage rumen fermentation and post-ruminal CP digestibility than storage temperature. Even though the effect of the interaction of silages and CP level on rumen fermentation and post-ruminal CP digestibility followed the same pattern as the interaction of fresh forages and CP level, it became evident that ensiling of forages until 75 d increased the end products of microbial fermentation in the rumen compared to fresh forages and prolonged storage beyond 75 d. Increasing the length of ensiling and CP of diets enhances CP ruminal degradation. However, ensiling beyond 75 d reduces CP digestibility to an extent that cannot be recovered by supplying additional CP. Finally, higher temperature of silage stored outdoors negatively influenced the CP intestinal digestibility.

## Figures and Tables

**Table 1 animals-12-03400-t001:** Chemical composition (in g/kg DM or as stated) of forages as affected by ensiling length and storage temperature.

Forage	Variable	Ensiling Lengths			Storage Temperature	
		0 d	75 d	180 d	25 °C	30 °C
	Dry matter	925	890	902	893	900
Sorghum	Organic matter	911	913	916	913	916
	Crude protein	99	96	96	99	94
	Neutral detergent fiber	506	418	496	451	465
	Acid detergent fiber	255	241	290	187	198
	ADFom (g/kg OM)	250	252	281	255	278
	Crude lipid	25	29	22	25	25
	ADIN	1.37	1.85	1.87	0.98	2.45
	Metabolizable energy (MJ/kg DM)	9.70	9.51	9.11	9.41	9.10
	Dry matter	924	924	925	920	928
Soybean	Organic matter	914	909	906	912	904
	Crude protein	148	170	155	166	160
	Neutral detergent fiber	412	416	507	439	485
	Acid detergent fiber	297	296	376	318	354
	ADFom (g/kg OM)	274	291	332	308	316
	Crude lipid	14	21	19	21	19
	ADIN	2.54	2.79	2.47	2.02	3.24
	Metabolizable energy (MJ/kg DM)	9.02	9.11	8.50	8.91	8.61

ADIN, Acid detergent insoluble nitrogen; ADFom, Acid detergent fiber after ashing.

**Table 2 animals-12-03400-t002:** Ingredient and chemical composition of the experimental diets at different crude protein levels for the in vitro fermentation.

Ensiling Length	0 d		75 d				180 d			
Storage Temperature			25 °C		30 °C		25 °C		30 °C	
Crude Protein Levels (g/kg DM)	90	130	90	130	90	130	90	130	90	130
Ingredient composition of diets (g/kg as fed basis)										
Sorghum	480	480	480	480	480	480	480	480	480	480
Soybean	320	320	320	320	320	320	320	320	320	320
Soy protein	0	100	0	100	0	100	0	100	0	100
Corn starch	200	100	200	100	200	100	200	100	200	100
Chemical composition of the diets (g/kg DM)										
Organic matter	929	921	926	919	931	924	932	925	927	920
Crude protein	92	132	96	136	92	132	92	132	88	129
Neutral detergent fiber	375	408	332	365	336	369	381	414	420	453
Acid detergent fiber	217	227	211	220	210	219	246	255	273	282
Crude lipid	16.5	17.0	20.5	21.0	20.5	21.0	17.2	17.7	16.1	16.6
Non-structural carbohydrates	446	364	478	397	483	402	442	361	403	321
ADIN	1.47	2.91	1.44	2.88	2.11	3.55	1.26	2.70	2.12	3.56
Metabolizable energy (MJ/kg DM)	9.53	9.65	9.55	9.68	9.38	9.50	9.15	9.27	9.00	9.12

DM, dry matter; ADIN, acid detergent insoluble nitrogen; soy protein (crude protein 434 g/kg DM, crude lipids 5.0 g/kg DM, crude ash 75.3 g/kg DM); corn starch (crude protein 2.0 g/kg DM, crude ash 1 g/kg DM).

**Table 3 animals-12-03400-t003:** Effect of ensiling length (L), storage temperature (T), and crude protein (CP) level on gas production (GP), pH, ammonium–N (NH_3_–N), and total ciliate protozoa count in rumen inoculum at different incubation periods (least squares means; n = 6).

Ensiling Length		0 d		75 d				180 d				SEM	*p*-Value				
Storage Temperature				25 °C		30 °C		25 °C		30 °C			Ensiling Length				
CP Levels (g/kg DM)		90	130	90	130	90	130	90	130	90	130		Lin	Quad	T	CP	L*CP
Variables	Time																
GP (mL/g DM)	8 h	77.5	67.0	88.8	64.1	91.6	80.6	71.3	57.4	68.6	58.2	4.06	<0.01	<0.01	n.s	<0.01	n.s
	24 h	151	149	176	132	174	160	157	132	150	141	8.38	n.s	0.02	n.s	<0.01	n.s
pH	8 h	6.74	6.73	6.73	6.72	6.73	6.73	6.74	6.74	6.74	6.74	0.01	n.s	n.s	n.s	n.s	n.s
	24 h	6.62	6.67	6.63	6.68	6.40	6.66	6.66	6.66	6.66	6.68	0.02	n.s	n.s	n.s	<0.01	n.s
NH_3_-N (mg/L)	8 h	20.6	25.5	26.4	27.5	27.4	28.3	26.0	25.7	23.7	25.3	1.28	n.s	<0.01	n.s	<0.01	0.04
	24 h	33.0	39.3	40.0	44.2	38.9	43.0	37.1	42.4	37.8	41.2	2.52	n.s	0.01	n.s	<0.01	n.s
Protozoa (×10^3^/mL)	8 h	6.21	5.38	3.85	2.27	3.23	3.48	3.17	2.10	4.74	4.00	1.06	<0.01	0.02	n.s	n.s	n.s
	24 h	7.71	5.93	5.84	3.86	5.60	4.96	5.14	4.33	4.55	4.02	1.02	<0.01	n.s	n.s	0.04	n.s

SEM, standard error of means; n.s, not significant. Lin, Linear; Quad, Quadratic; L*CP, interaction effects of ensiling length with crude protein level. L*T, interaction effects of ensiling length with storage temperature (*p* > 0.1). CP*T, interaction effects of crude protein level with storage temperature (*p >* 0.1). L*CP*T, interaction effects between ensiling length, crude protein level and storage temperature (*p* > 0.1).

**Table 4 animals-12-03400-t004:** Effect of ensiling length (L), storage temperature (T), and crude protein (CP) level on the in vitro fermentation of short-chain fatty acid (SCFA) concentration and individual SCFA proportions at different incubation periods (least squares means; n = 6).

Ensiling Length		0		75 d				180 d				SEM	*p*-Value				
Storage Temperature				25 °C		30 °C		25 °C		30 °C			Ensiling Length				
CP Levels (g/kg DM)		90	130	90	130	90	130	90	130	90	130		Lin	Quad	T	CP	L*CP
Variables	Time																
Total SCFA ^1^ (µmol/mL)	8 h	29.6	29.6	32.6	31.9	32.0	31.6	31.6	30.4	29.9	29.1	0.57	n.s	<0.01	0.06	n.s	n.s
	24 h	44.6	44.0	47.0	45.8	47.0	46.2	45.7	44.1	44.6	44.0	0.93	n.s	<0.01	n.s	0.08	n.s
Individual SCFA proportions (µmol/100 µmol total SCFA)																	
Acetate (C2)	8 h	68.2	69.1	64.4	64.7	65.0	65.6	65.1	65.7	65.5	66.3	0.42	<0.01	<0.01	n.s	<0.01	n.s
	24 h	65.5	65.8	63.7	62.0	64.0	62.3	64.3	62.4	64.5	62.6	0.56	<0.01	<0.01	n.s	<0.01	<0.05
Propionate (C3)	8 h	17.5	17.2	19.2	21.3	19.0	20.8	18.7	20.4	18.5	20.2	0.68	<0.01	<0.01	n.s	<0.01	<0.05
	24 h	18.4	18.4	19.6	21.6	19.0	21.4	19.3	21.2	19.3	21.1	0.42	<0.01	<0.01	n.s	<0.01	<0.01
Iso-butyrate	8 h	0.91	0.90	1.02	0.91	1.00	0.89	1.03	0.92	1.03	0.92	0.04	<0.05	n.s	n.s	<0.01	0.08
	24 h	0.99	1.01	1.07	1.03	1.10	1.04	1.07	1.06	1.04	1.04	0.02	<0.05	<0.05	n.s	n.s	n.s
Butyrate (C4)	8 h	11.6	11.0	13.1	10.9	13.0	10.7	12.9	10.9	12.8	10.6	0.39	n.s	n.s	n.s	<0.01	<0.01
	24 h	12.6	12.3	12.6	12.6	13.0	12.5	12.3	12.6	12.3	12.5	0.33	n.s	n.s	n.s	n.s	n.s
Iso-valerate	8 h	0.93	0.92	1.21	1.01	1.20	0.96	1.24	1.02	1.20	0.97	0.08	<0.01	<0.05	n.s	<0.01	0.07
	24 h	1.45	1.48	1.68	1.55	1.70	1.54	1.67	1.55	1.59	1.53	0.08	n.s	n.s	n.s	n.s	n.s
Valerate	8 h	0.85	0.88	1.03	1.14	1.00	1.09	0.99	1.08	0.97	1.04	0.02	<0.01	<0.01	<0.05	<0.01	n.s
	24 h	1.06	1.12	1.32	1.23	1.30	1.22	1.27	1.19	1.23	1.17	0.03	<0.01	<0.01	n.s	<0.05	<0.01
C2:C3	8 h	3.89	4.05	3.39	3.04	3.48	3.17	3.52	3.23	3.58	3.29	0.29	<0.01	<0.01	n.s	n.s	n.s
	24 h	3.60	3.60	3.25	2.87	3.35	2.91	3.34	2.94	3.35	2.97	0.34	<0.01	<0.01	n.s	<0.01	<0.05
(C2+C4):C3	8 h	4.62	4.70	4.07	3.55	4.18	3.67	4.22	3.76	4.29	3.82	0.38	<0.01	<0.01	n.s	<0.01	0.05
	24 h	4.29	4.26	3.89	3.46	3.99	3.50	3.97	3.54	3.99	3.56	0.37	<0.01	<0.01	n.s	<0.01	<0.05
Total BCFA	8 h	1.83	1.82	2.23	1.92	2.30	1.84	2.27	1.94	2.24	1.89	0.12	<0.01	n.s	n.s	<0.01	n.s
	24 h	2.45	2.49	2.76	2.58	2.70	2.59	2.74	2.61	2.63	2.57	0.10	<0.05	n.s	n.s	n.s	n.s

^1^ Total SCFA corrected for the SCFA concentrations in the inoculum of the blank bottle at each incubation sampling time. C2:C3, Acetate: Propionate; (C2 + C4):C3, Acetate + Butyrate: Propionate, BCFA, branched chain fatty acids (isobutyrate + isovalerate). Lin, Linear; Quad, Quadratic; SEM, standard error of means; n.s, not significant. L*T, interaction effects of ensiling length with storage temperature (*p* > 0.1), CP*T, interaction effects of crude protein level with storage temperature (*p >* 0.1). L*CP*T, interaction effects between ensiling length, crude protein level and storage temperature (*p* > 0.1).

**Table 5 animals-12-03400-t005:** Effect of ensiling length, storage temperature, and crude protein level on the in vitro degradability and post-ruminal digestibility of diets’ dry matter (DM) and crude protein (CP) at different incubation periods (least squares means; n = 6).

Ensiling Length		0 d		75 d				180 d				SEM	*p*-Value				
Storage Temperature				25 °C		30 °C		25 °C		30 °C			Ensiling Length				
CP Levels (g/kg DM)		90	130	90	130	90	130	90	130	90	130		Lin	Quad	T	CP	L*T
Variables	Time																
Apparent ruminal DM degradability (g/100 g)	8 h	40.7	42.8	45.8	46.6	42.9	45.6	47.2	45.6	45.3	44.7	4.52	n.s	n.s	n.s	n.s	n.s
	24 h	49.7	48.7	50.2	48.2	50.3	48.9	50.6	45.7	46.7	45.7	1.76	0.09	n.s	n.s	n.s	n.s
Apparent intestinal DM digestibility (g/100 g)	8 h	48.3	47.9	48.3	48.6	48.8	50.0	46.1	46.5	45.4	44.0	0.85	<0.01	<0.01	n.s	n.s	n.s
	24 h	45.0	46.8	45.7	45.9	43.9	44.8	41.8	38.8	38.7	39.1	1.04	<0.01	<0.01	n.s	n.s	n.s
Apparent total DM digestibility (g/100 g)	8 h	66.0	67.1	67.4	68.2	67.2	68.8	67.0	66.9	64.7	65.0	2.91	n.s	n.s	n.s	n.s	n.s
	24 h	72.3	72.7	72.9	71.9	72.1	71.7	71.2	66.8	67.3	66.8	0.97	<0.01	0.01	n.s	n.s	n.s
CP in undegraded substrate (mg CP)	8 h	205	235	184	213	192	211	150	187	146	186	12.5	<0.01	n.s	n.s	<0.01	n.s
	24 h	179	208	166	206	172	202	142	186	148	178	6.14	<0.01	n.s	n.s	<0.01	n.s
Apparent intestinal CP digestibility (g/100 g)	8 h	71.2	74.9	73.0	74.7	75.9	76.9	76.2	77.4	73.5	75.1	1.26	<0.01	n.s	n.s	<0.01	0.02
	24 h	74.2	75.5	70.9	72.4	70.8	72.0	71.3	71.9	66.8	66.5	0.80	<0.01	0.02	<0.01	<0.01	<0.01

SEM, standard error of means; n.s, not significant. Lin, Linear; Quad, Quadratic; L, ensiling length; T, storage temperature. L*CP, interaction effects of ensiling length with crude protein level (*p* > 0.1), CP*T, interaction effects of crude protein level with storage temperature (*p >* 0.1). L*CP*T, interaction effects between ensiling length, crude protein level and storage temperature (*p* > 0.1).

## Data Availability

The datasets analyzed are available from the corresponding author on request.

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
