# Peer review of "In Vitro Rumen Fermentation and Post-Ruminal Digestibility of Sorghum–Soybean Forage as Affected by Ensiling Length, Storage Temperature, and Its Interactions with Crude Protein Levels"

_animals, 2022, doi:10.3390/ani12233400_

Round 1

Reviewer 1 Report

In general, its of practical importance even in tropical and sub tropical climatic situations. Abstract may be simplified to some extent for general audiences. outdoor and indoor temp may be mentioned in the relevant part, pointed out in the text. 

Author Response

Point 1: Outdoor and indoor temperatures may be mentioned in the relevant part pointed out in the text.

Response 1: The temperatures have been mentioned as suggested in line 102 - 106

Reviewer 2 Report

The study is an interesting one as it has very strong practical application in the tropical context. Thes study design is good and the results adequately presented. The discussion provides explanation and justification to results obtained. However, the study result section can be improved especially in the interpretation of interaction effects. 

General comments

Change the length to duration for the ensiling period.

The result should be more specifically narrated especially in areas of the interaction effect.

Use the MDPI reference format throughout the manuscript. References should be in chronological order.

Two major issues justify this study, a) prolonged dry seasons in the tropical regions which warrant prolonged ensiling of feed. b) High temperatures are associated with tropical environments and hence affect ensiling conditions which are in contrast to temperate regions. Authors must exemplify these two key issues in the opening or closing paragraphs of the introduction as well as the discussion.

Specific Objectives

Title: change ensiling length to ensiling duration and across the manuscript.

L14: sentence correction. ‘purpose of study’

L15: sorghum-soybean forage mixtures

L16: post-ruminal nutrient digestibility

L17: change prolonged ensiling beyond 75 d to ‘forage ensiled beyond 75 d’

L20/21: delete ‘to the extent that…’. Just say significantly.

L22: sentence correction. ‘aim of the study’. Ensiling duration

L23: sorghum-soybean forage mixtures

L24: digestibility of nutrients.

L25: comma after 180 d,

L31: increases= increased

L32: increased quadratically with ensiling duration.

L31-33: Were interaction effects non-significant for these parameters?

What does this interaction effect say? This should be clearly stated.

L44-45: reduced CP% and additional CP inclusion did not ameliorate this.

L51: ‘with an expected net feeding strategy’. The intent of the authors is not clearly communicated here. Recast.

L64: sheep consuming fresh ..

L65: sorghum-soybean forage? Always specify when it is forage as these plants also produced grains which could be fed in another context.

L71: In view of the impacts of ensiling conditions such as ….

L80-83: sentence needs corrections. Too many thoughts but disjointed.

Grass silage containing tropical legumes resulted in higher performance (average daily gain, milk yield) than those without tropical legume inclusion and it is assumed that forage legume CP appeared to be more digestible post-ruminally. Is the the thought?

How does this reconcile with more rapid degradation of ensiled legume in the rumen which often result in excess DP and rumen ammonia?

L109: delete ‘Hence,’

L110: delete ‘in the present experiment’

L125: remove space before Menke

L127: Change diet ingredient to ‘Each substrate’ What quantity of each ingredient made the substrate? Silage plus soyprotein/starch? Provide more detail quantity for each treatment. Reffer to table 2.

 Sampling

Sampling should contain gas measurement. .before ‘at the end of each incubation’…

Merge the paragraph in L142-145 to the preceding paragraph.

Table 1

Indoor vs outdoor, although reflective but not a clear description of temperature. Outdoor is often colder in winter and hotter in most tropical regions all year round. Can you specify average temperature (in all tables and in methodology section).

Chemical analysis

Which nutrients were analysed and which were calculated? Specify.

Table 2: This was all calculated or analysed?

For Sorghum and Soybean, add ‘silage’

L194: define PPS at first use.

L212: how does L212-213 differ from L214-215. Once CP is rumen undegradable and also intestinally undigestible, that should be ADIN. Please harmonise.

L269: To what extent was this valid? Where there are no interaction effects, authors ought to present results of simple main effects- storage duration, temperature, or CP level. Where thereis interaction, the results are discussed within each sub-factor explaining what patterns or shift in values were observed.

Results

Table 3

Because this is a 3-factor experimental layout, the 3-way interaction effect should be the first focus. The authors must document this. If there are no sig. 3-way interaction, this should be stated and the authors can then justify why results of simple main effects as well as 2 way interaction of CP and duration was presented. T*CP*Lengt= NS?; T*Length= NS?; T*CP= NS? L*CP is presented and only NH4 at 8h incubation showed significance. This can then justifiably allow the authors to discuss the simple main effects of Length of storage (linear and Quadratic) as well as simple main effect of Temperature for the other parameters.

For ease of interpretation, I would suggest the tables are presented as 2 way interactions only while any significant 3 way interaction are mentioned.

Here is a suggestion:

The interactive description of propionate can be written similar to this : While propionate proportion was not affected by CP level at o day of ensiling, propionate proportion was higher in the higher CP silage in the ensiled forage sampled at 75 and 180 days.

Something similar to this can be adopted across the narration of result for subsequent descriptions.

L275277: affected is a double-edged sword. Simply state that ensiling duration resulted in linear and quadratically increase in GP to xxx and then reduced subsequently while higher CP resulted in lower GP irrespective of storage duration.

Table 4

Under variables, simple write Acetate (C2; Propionate (C3) butyrate (C4).

The acetate to propionate ratio need to be looked at as the values do not seem correct.

For example, acetate at 24 h; 0 day and 90 CP is 65.5 and propionate is 18.4. acetate to propionate will be 65.5/18.4= 3.59. The values across the table does not align with this range.

Discussion

Ammonia is also a major end product, although used up by micorbes, a considerable portion is either recycled or lost.

L404: 75 d length may actually not be as the actual inflexion point is unknown. Better say around 75 d or near 75 day since it is a curve.

L410-411: ‘it appears that energy supply and 410 availability of ruminal N increased microbial growth and activities which positively influences GP’…since the increase in CP did not improve  GP, what we may infer is that energy becomes limiting beyond that stage- around 75 d. If energy had been higher, increased CP would have resulted in increased microbial growth manifesting in higher GP.

L416-417: Moreover, the fibre concentrations of silages stored for 75 d may be more degradable, indicating lower usage of readily fermentable carbohydrates by silage microbes dur ing ensiling at 75 d compared to 180 d’…180 d ensiling depleted fermentable carbs in the silage. This you have stated in L430.

L435-439: sentences need to be re-casted. Some of the narrations are at best ‘result’ rather than discussion. You can reduce these sentences across the discussion and make it simpler and shorter.

L450:sentence correction. Recast as ‘consistent with higher protozoal counts in inoculum from diets with lower CP concentration in the present study. Equally, there is a positive correlation between protozoal population with increased starch concentration (Dijkstra, 1994) and this was observable in the current study. Furthermore, starch is an essential….

L457: increasing fibre proportion. In my opinion, what changed is higher proportion of fibre as a result of depleted easily fermentable portion of the feed- NSC, CP etc.

L461-463: prolonged fermentation time affords sustained microbial degradation. Do not forget that diet didn’t change from 8- 24h but ammonia increased as a result of sustained fermentation.

L470: change finding to observation

L480: change amount to concentration…with increasing length, suggesting higher….

L480-484: Break the sentence into 2.

L503: Don’t start paragraph with ‘On the other hand’.

L503-507: Lima et al observed a positive effect of ensiling on sorghum-soybean forage mixtures. In that study, forage ensiled between 162- 182 d showed higher proportion of intestinal digestible CP although the authors did provide details of the causal factors.

Conclusions

Simply state:

Ensiling duration had a greater impact on silage rumen fermentation and post0-ruminal CP digestibility than storage temperature.

L523: interaction effect of silage and CP level??? There is a confusion here. Silage duration or temperature?

Silages ensiled up to 75 d resulted in higher fermentation products- SCFA, GP??, than the fresh forage or the 120 d silage samples.

What is outdoor? Higher temperature? Be specific.

Author Response

Title: change ensiling length to ensiling duration and across the manuscript.

Response: We humbly decline the suggestion because the study is a continuation of a previous silage study with similar studied factor at different level. Also, ensiling length is often used in silage studies than duration.

L14: sentence correction. ‘purpose of study’

Response: corrected as suggested

Response: corrected as suggested

L15: sorghum-soybean forage mixtures

Response: corrected as suggested

L16: post-ruminal nutrient digestibility

Response: corrected as suggested

L17: change prolonged ensiling beyond 75 d to ‘forage ensiled beyond 75 d’

Response: corrected as suggested

L20/21: delete ‘to the extent that…’. Just say significantly.

Response: corrected as suggested

L22: sentence correction. ‘aim of the study’. Ensiling duration

Response: corrected as suggested

L23: sorghum-soybean forage mixtures

Response: corrected as suggested

L24: digestibility of nutrients.

Response: corrected as suggested

L25: comma after 180 d,

Response: corrected as suggested

L31: increases= increased

Response: corrected as suggested

L32: increased quadratically with ensiling duration.

Response: corrected as suggested

L31-33: Were interaction effects non-significant for these parameters?

Response: corrected as suggested

What does this interaction effect say? This should be clearly stated.

Response: It has been stated as suggested

L44-45: reduced CP% and additional CP inclusion did not ameliorate this.

Response: corrected as suggested

L51: ‘with an expected net feeding strategy’. The intent of the authors is not clearly communicated here. Recast.

L64: sheep consuming fresh ..

Response: corrected as suggested

L65: sorghum-soybean forage? Always specify when it is forage as these plants also produced grains which could be fed in another context.

Response: corrected as suggested

L71: In view of the impacts of ensiling conditions such as ….

Response: corrected as suggested

L80-83: sentence needs corrections. Too many thoughts but disjointed.

Grass silage containing tropical legumes resulted in higher performance (average daily gain, milk yield) than those without tropical legume inclusion and it is assumed that forage legume CP appeared to be more digestible post-ruminally. Is the the thought?

Response: corrected as suggested

How does this reconcile with more rapid degradation of ensiled legume in the rumen which often result in excess DP and rumen ammonia?

Response: The rapid degradation of silage CP in the rumen may render CP unavailable post ruminally and this is dependent on forage properties as well as the ensiling conditions

L109: delete ‘Hence,’

Response: corrected as suggested

L110: delete ‘in the present experiment’

Response: deleted as suggested

L125: remove space before Menke

Response: corrected as suggested

L127: Change diet ingredient to ‘Each substrate’ What quantity of each ingredient made the substrate? Silage plus soyprotein/starch? Provide more detail quantity for each treatment. Reffer to table 2.

Response: corrected as suggested and more details about the quantity of individual ingredients have been included in the material and method section

 Sampling

Sampling should contain gas measurement. .before ‘at the end of each incubation’…

Response: corrected as suggested

Merge the paragraph in L142-145 to the preceding paragraph.

Response: corrected as suggested

Table 1

Indoor vs outdoor, although reflective but not a clear description of temperature. Outdoor is often colder in winter and hotter in most tropical regions all year round. Can you specify average temperature (in all tables and in methodology section).

Response: The temperature has been specified as suggested

Chemical analysis

Which nutrients were analysed and which were calculated? Specify.

Response: nutrients analysed were well specified as well those calculated. The calculation section highlights the calculated nutrients

Table 2: This was all calculated or analysed?

Response: Both

For Sorghum and Soybean, add ‘silage’

Response: The ingredient is composed of both fresh forage and silage. It will be misleading to add silage

L194: define PPS at first use.

Response: defined as suggested

L212: how does L212-213 differ from L214-215. Once CP is rumen undegradable and also intestinally undigestible, that should be ADIN. Please harmonise.

Response: harmonized as suggested

L269: To what extent was this valid? Where there are no interaction effects, authors ought to present results of simple main effects- storage duration, temperature, or CP level. Where there is interaction, the results are discussed within each sub-factor explaining what patterns or shift in values were observed.

Response: We believe our explanation does not deviate from your suggestion. It was categorically stated in the statistical analysis section that where there were no interactions, no report would be provided.  Also, the results were discussed for each studied factor accordingly.

Results

Table 3

Because this is a 3-factor experimental layout, the 3-way interaction effect should be the first focus. The authors must document this. If there are no sig. 3-way interaction, this should be stated and the authors can then justify why results of simple main effects as well as 2 way interaction of CP and duration was presented. T*CP*Lengt= NS?; T*Length= NS?; T*CP= NS? L*CP is presented and only NH4 at 8h incubation showed significance. This can then justifiably allow the authors to discuss the simple main effects of Length of storage (linear and Quadratic) as well as simple main effect of Temperature for the other parameters.

For ease of interpretation, I would suggest the tables are presented as 2 way interactions only while any significant 3 way interaction are mentioned.

Response: We believe the table was presented in a 2-way interaction manner with the exclusion of 2-way interactions that were not significant across all variables. The footnote below each table also explain further the exclusion of the those interactions.

Here is a suggestion:

The interactive description of propionate can be written similar to this : While propionate proportion was not affected by CP level at o day of ensiling, propionate proportion was higher in the higher CP silage in the ensiled forage sampled at 75 and 180 days.

Response: This suggestion has been adopted 

Something similar to this can be adopted across the narration of result for subsequent descriptions.

L275277: affected is a double-edged sword. Simply state that ensiling duration resulted in linear and quadratically increase in GP to xxx and then reduced subsequently while higher CP resulted in lower GP irrespective of storage duration.

Response: corrected as suggested

Table 4

Under variables, simple write Acetate (C2; Propionate (C3) butyrate (C4).

Response: corrected as suggested

The acetate to propionate ratio need to be looked at as the values do not seem correct.

For example, acetate at 24 h; 0 day and 90 CP is 65.5 and propionate is 18.4. acetate to propionate will be 65.5/18.4= 3.59. The values across the table does not align with this range.

Response: The values have been corrected as suggested, and the statistical analysis was not affected by the correction.

Discussion

Ammonia is also a major end product, although used up by micorbes, a considerable portion is either recycled or lost.

Response: Ammonia has been included in the major end product of rumen fermentation as suggested

L404: 75 d length may actually not be as the actual inflexion point is unknown. Better say around 75 d or near 75 day since it is a curve.

Response: corrected as suggested

L410-411: ‘it appears that energy supply and 410 availability of ruminal N increased microbial growth and activities which positively influences GP’…since the increase in CP did not improve  GP, what we may infer is that energy becomes limiting beyond that stage- around 75 d. If energy had been higher, increased CP would have resulted in increased microbial growth manifesting in higher GP.

Response: The suggestion have been implemented

L416-417: Moreover, the fibre concentrations of silages stored for 75 d may be more degradable, indicating lower usage of readily fermentable carbohydrates by silage microbes dur ing ensiling at 75 d compared to 180 d’…180 d ensiling depleted fermentable carbs in the silage. This you have stated in L430.

Response: 

L435-439: sentences need to be re-casted. Some of the narrations are at best ‘result’ rather than discussion. You can reduce these sentences across the discussion and make it simpler and shorter.

Response: The sentence have been made simpler and shorter as suggested

L450:sentence correction. Recast as ‘consistent with higher protozoal counts in inoculum from diets with lower CP concentration in the present study. Equally, there is a positive correlation between protozoal population with increased starch concentration (Dijkstra, 1994) and this was observable in the current study. Furthermore, starch is an essential….

Response: The sentence have been corrected

L457: increasing fibre proportion. In my opinion, what changed is higher proportion of fibre as a result of depleted easily fermentable portion of the feed- NSC, CP etc.

Response: corrected as suggested

L461-463: prolonged fermentation time affords sustained microbial degradation. Do not forget that diet didn’t change from 8- 24h but ammonia increased as a result of sustained fermentation.

Response: Fitly said and what was stated in these lines did not deviate from the opinion given

L470: change finding to observation

Response: changed as suggested

L480: change amount to concentration…with increasing length, suggesting higher….

Response: changed as suggested

L480-484: Break the sentence into 2.

Response: corrected as suggested

L503: Don’t start paragraph with ‘On the other hand’.

Response: corrected as suggested

L503-507: Lima et al observed a positive effect of ensiling on sorghum-soybean forage mixtures. In that study, forage ensiled between 162- 182 d showed higher proportion of intestinal digestible CP although the authors did provide details of the causal factors.

Response: The suggested version have been included

Conclusions

Simply state:

Ensiling duration had a greater impact on silage rumen fermentation and post0-ruminal CP digestibility than storage temperature.

Response: corrected as suggested

L523: interaction effect of silage and CP level??? There is a confusion here. Silage duration or temperature?

Response: Our belief is that the conclusion section states the findings that are not obvious. The reference is to ensiling duration. The fresh forage (0 d) and silage (either at 75 or 180 d) interaction with CP levels have a similar pattern.

Silages ensiled up to 75 d resulted in higher fermentation products- SCFA, GP??, than the fresh forage or the 120 d silage samples.

What is outdoor? Higher temperature? Be specific.

Response: corrected as suggested

Reviewer 3 Report

Dear Authors,

One of the main issues facing nutritionists and animal feed technicians is to improve forage preservation processes. This manuscript addresses this issue, aiming to evaluate the effects of silage length, storage temperature, and their interaction with crude protein levels in sorghum-soybean on in vitro rumen fermentation and post-rumen digestibility.

At a time when crude protein in feed is extremely valued, all data on this issue are important contributions to scientific knowledge and with practicability to producers. Given this, the article submitted for review is current and appropriate for Animal magazine. 

The article has a clear and well defined main objective - To evaluate the in vitro ruminal fermentation and post-ruminal digestibility of diets containing combination of fresh or ensiled sorghum and soybean forage, affected by length of silage, storage temperature and their interaction with dietary BW levels, thus we can know the potential of this green and ensiled forage comparing different factors with the length of silage, important for rumination and consequently for VFA and milk solids content.

The abstract and introduction are simple and well written, it seeks to justify the topic of the paper and the importance of crude protein and how it is degraded in different types of forage using studies to justify. The introduction could be improved if it characterizes the forages under study (sorghum). In my opinion the introduction lacks more literature references to justify some statements.

In the section materials and methods are written in a way that the essay is replicable and well presented. It could include, if it has possibility the lignin content (ADL), it can be fundamental for the particle size of the silage.

In Table 1 - they indicate that the units are expressed in g/kg DM, in the table should be indicated the parameter DM content. 

Table 2 is the same as table 1.

L187 - Shouldn't the temperature be 38ºC? 

To determine NSC it was necessary to determine the aNDF, this value is not referenced in the food tables. How was it determined?

The same happens with the ADFform, you must include the contents in the tables and indicate the method used to determine it.

Overall, the results are presented correctly, in 3 tables. Each parameter analysed is described in detail. They should indicate the average temperatures Indoor/outdoor.

The discussion made is very well done, as the most important and real relationships between the already established controlled parameters are also indicated here. In this sense, the results are confirmatory. However, in the discussion old references were used to justify the data, only one reference, used 2 times, from the year 2022.

The conclusion made reflects the results obtained.

The importance of the content is assessed as medium, as the study is only part of the research - further studies are needed to characterise these forages, for example the final practical nutritional effects (positive, adverse?) in animal trials.

Author Response

In the section materials and methods are written in a way that the essay is replicable and well presented. It could include, if it has possibility the lignin content (ADL), it can be fundamental for the particle size of the silage.

Response: The lignin concentration was not analysed, instead the ADIN concentration was analysed. Since there is a correlation between both we believe the ADIN concentration can provide the necessary information as presented

In Table 1 - they indicate that the units are expressed in g/kg DM, in the table should be indicated the parameter DM content. 

Response: The DM content has been included as suggested

Table 2 is the same as table 1.

Response: Table 1 shows the chemical composition of the forages based on lab analysis whereas Table 2 shows the ingredient composition of the substrate diet and the chemical composition

L187 - Shouldn't the temperature be 38ºC? 

Response: Typographical error. It has been changed

To determine NSC it was necessary to determine the aNDF, this value is not referenced in the food tables. How was it determined?

Response: The value of NDF was provided in both table

The same happens with the ADFform, you must include the contents in the tables and indicate the method used to determine it.

Response: The method and the ADFom value have been included

Overall, the results are presented correctly, in 3 tables. Each parameter analysed is described in detail. They should indicate the average temperatures Indoor/outdoor.

The discussion made is very well done, as the most important and real relationships between the already established controlled parameters are also indicated here. In this sense, the results are confirmatory. However, in the discussion old references were used to justify the data, only one reference, used 2 times, from the year 2022.

Response: Fitly said. The focus was partly on legume silage in the tropics but amendment has been made with the inclusion of another recent reference

The conclusion made reflects the results obtained.

The importance of the content is assessed as medium, as the study is only part of the research - further studies are needed to characterise these forages, for example the final practical nutritional effects (positive, adverse?) in animal trials.